

# Peptidomics insights: neutrophil extracellular traps (NETs) related to the chronic subdural hemorrhage

Jie Li[1,*], Jing Zhao[2,*], Shuchen Sun[1], Sen Shen[1], Bincheng Zhong[3] and Xiaohua Dong[1,2]

[1] Department of Neurosurgery, Tongren Hospital, Shanghai Jiao Tong University School of Medicine, Shanghai, China

[2] Hongqiao International Institute of Medicine, Tongren Hospital, Shanghai Jiao Tong University School of Medicine, Shanghai, China

[3] Department of Emergency, Tongren Hospital, Shanghai Jiao Tong University School of Medicine, Shanghai, China

[*] These authors contributed equally to this work.

## ABSTRACT

Chronic subdural hemorrhage (CSDH) refers to a hematoma with an envelope between the dura mater and the arachnoid membrane and is more common among the elderly. It was reported that the dura mater, which is highly vascularized with capillary beds, precapillary arterioles and postcapillary venules play an important role in the protection of the central nervous system (CNS). Numerous evidences suggests that peptides play an important role in neuroprotection of CNS. However, whether dura mater derived endogenous peptides participate in the pathogenesis of CSDH remains undetermined. In the current study, the peptidomic profiles were performed in human dura of CSDH (three patients) and the relative control group (three non-CSDH samples) by LC-MS (liquid chromatography–mass spectrometry). The results suggested that a total of 569 peptides were differentially expressed in the dura matter of CSDH compared with relative controls, including 217 up-regulated peptides and 352 down-regulated peptides. Gene Ontology (GO) analysis demonstrated that the precursor proteins of those differentially expressed peptides were involved in the various biological processes. Interestingly, Kyoto Encyclopedia of Genes and Genomes (KEGG) analysis suggested that NETs participated in the pathogenies of CSDH. Further investigate showed that H3Cit was significantly elevated in the dural and hematoma membranes of patients with CSDH compared to patients without CSDH. Taken together, our results showed the differentially expressed peptides in human dura mater of CSDH and demonstrated that NETs formation in the dural and hematoma membranes might be involved in the pathogenesis of CSDH. It is worth noting that pharmacological inhibition of NETs may have potential therapeutic implications for CSDH.

Corresponding author
Xiaohua Dong,
njfishxiaohua@163.com

# INTRODUCTION

Chronic subdural hemorrhage (CSDH) is a complex and common neurosurgical disease, which refers to patients with intracranial hemorrhage occurring in the subdural space
more than three weeks (*Shrestha et al., 2022*; *Chen & Levy, 2000*). The incidence of CSDH ranges from 1 to 20 per 100,000 people per year and is on the rise with the increasing aging of the population worldwide (*Hamou et al., 2022*; *Feghali, Yang & Huang, 2020*). In general, patients with asymptomatic CSDH are generally treated conservatively, including the intracranial pressure control, medical treatment, and anticoagulant reversal (*Feghali, Yang & Huang, 2020*). For symptomatic patients, surgery operation is often the preferred treatment, such as burr hole craniostomy, twist drill craniostomy, and middle meningeal artery embolization (*Ban et al., 2018*). However, recurrence rates of CSDH after surgery varied from 2% to 37% (*Guo et al., 2022*). Therefore, it is particularly urgent to find a safety and potentially preventative treatment option for CSDH to avoid subjecting patients to multiple surgeries for CSDH.

Recent evidence suggests that multiple biological processes such as angiogenesis, inflammatory responses, and fibrinolysis are involved in promoting CSDH expansion (*Edlmann et al., 2017*). Local inflammatory response is the pathogenic processes underlying CSDH development (*Kitazono et al., 2012*; *Tang, Ai & Macdonald, 2011*). The proinflammatory cytokines were shown to be significantly raised in CSDH, such as IL-6 and IL-8 (*Suzuki et al., 1998*; *Bounajem et al., 2021*). In addition, pro-inflammatory mediators such as tissue necrosis factor $\alpha$ (TNF-$\alpha$), vascular endothelial growth factor (VEGF), and matrix metalloproteinases promote the formation of immune, fragile and leaky blood vessels, promote chronic bleeding, and play an important role in the formation of CSDH (*Bounajem et al., 2021*). Therefore, anti-inflammatory drug therapy is considered to be potentially beneficial for patients with CSDH (*Vychopen, Guresir & Wach, 2022*). It was reported that treatment with atorvastatin which has an anti-inflammatory function could improve the neurological functions in patients with CSDH (*He et al., 2021*; *Araujo et al., 2010*). However, whether other proinflammatory biological processes are involved in the pathogenesis of CSDH remains to be determined.

Peptidomics is a discipline that studies the structure, function, variation and correlation of endogenous peptides and low molecular weight proteins. Peptides are a general class of chemical substances consisting of amino acids residing in living organisms. Until now, numerous peptides have been found in living organisms and have a wide range of physiological activities. AnxA1 N-terminal derived peptide Ac2-26 play an important role in regulate the inflammatory response in several disorders (*Ferreira et al., 2022*). This suggests that it is a new perspective to understand and study the occurrence, development and therapy of CSDH diseases from the perspective of peptidomics.

The dura, located between the skull and the brain, is the outermost layer of the meninges and is mainly composed of fibroblasts and collagen (*Weller et al., 2018*). The dura mater is a barrier to the brain's internal environment and is important for protecting and stabilizing the central nervous system (*de Oliveira et al., 2022*; *Adeeb et al., 2012*). Previously reported suggested that membranous structures on the inner surface of the dural membrane are the source of inflammation for hematoma formation (*de Oliveira et al., 2022*). In addition, the vascular system that exists in the human dura plays a crucial role in pathological diseases such as dural hematoma and meningitis (*Mecheri, Paris & Lubbert, 2018*). However,

whether the peptides derived from human dura mater involved in the pathogenesis of CSDH has not been elucidated.

In this study, a total of 569 differential peptides were identified in the CSDH patients' dura mater compared with dural of non-CSDH. Among the top dysregulated peptides, 10 down-regulated peptides were derived from Vimentin (VIM), and three up-regulated peptides were derived from fibrinogen. Recent study has uncovered the important role for VIM and fibrinogen in the progression of numerous neurological diseases, suggesting that these dysregulated peptides might participated in the progression of CSDH. In addition, KEGG and Western blot analysis revealed that NETs might play an important role in the pathogenesis of CSDH. These results suggest that different expressed peptides may be involved in the pathological process of CSDH, which provides a new insight into the pathogenesis of CSDH.

## MATERIALS & METHODS

### Sample collection

Dural tissues were collected from the patients with CSDH or cerebral hemorrhage (relative control group) at the Shanghai Tongren Hospital affiliated to Shanghai Jiao Tong University School of Medicine from October 2022 to February 2023. The total of six dural tissue were enrolled in the peptidomics analysis, which include three CSDH samples and three relative control samples. For Western blot analysis, the human dura mater and hematoma membranes were collected from another three CSDH patients, the relative control human dura mater were obtained from another three patients without CSDH. The clinical characteristics of the CSDH group and the control group are shown in Table 1. The inclusion criteria for clinical samples in the CSDH group were patients with chronic subdural hematoma who needed meninges trimmed after surgery, the inclusion criteria for clinical samples in the control group were patients with cerebral hemorrhage and cerebral hernia who needed meninges trimmed after surgery. The exclusion criteria are as follows, no previous history of cognitive impairment, cerebral infarction, brain trauma, stroke, mental disorders, or other brain disease, and no history of hepatitis, tuberculosis, typhoid, and other infectious diseases. We declare that we have received the written informed consent from participants in this study. This sample collection was approved by the ethical committee of Shanghai Tongren hospital (2022-078-01).

### Peptide extraction

The tissue samples were ground into powder by adding liquid nitrogen, and Tris-HCL was added according to the volume ratio of 1:3. The samples were heated and boiled for 10 min. The samples were crushed by ultrasound at 100 HZ in an ice water bath, over 5 s, at an interval of 5 s, for 2 min. Then, the final concentration of 1M glacial acetic acid was added into the sample tube and the sample was oscillated for 2 min. Add the final concentration of about 50% acetonitrile. After centrifugation at high speed at 12,000 g at 4 °C for 10 min, the supernatant was taken and transferred to a clean EP tube for freeze-drying. Add 80% acetone solution, vortex, oscillation, water bath ultrasonic for 2 min, 4 °C, 20,000 g high speed centrifugation for 30 min, take the supernatant, transfer to a clean EP tube, freeze
**Table 1  Baseline characteristics ($n = 12$).**

| Characteristics | Control | CSDH |
|---|---|---|
| Number | 6 | 6 |
| Age (mean ± SD, range) | 67.5 ± 6.5, 55–74 | 70 ± 11.2, 48–82 |
| Sex | | |
| Male, $n$ (%) | 4 (66.67%) | 6 (100%) |
| Female, $n$ (%) | 2 (33.33%) | 0 (0%) |
| GCS | | |
| 15, $n$ (%) | 5 (83.3%) | 0 (0%) |
| 13-15, $n$ (%) | 1 (16.6%) | 2 (33.3%) |
| 9-12, $n$ (%) | 0 (0%) | 4 (66.6%) |
| NIHSS | | |
| 1–4, $n$ (%) | 5 (83.3%) | 0 (0%) |
| 5–15, $n$ (%) | 1 (16.6%) | 5 (83.3%) |
| 15–20, $n$ (%) | 0 (0%) | 1 (16.6%) |
| Systolic pressure (mmHg) (mean ± SD, range) | 169.8 ± 15.5, 144–188 | 137.3 ± 6.18, 127–145 |
| Diastolic pressure (mmHg) (mean ± SD, range) | 98.3 ± 8.8, 96–111 | 82.5 ± 6.8, 74–94 |

drying. A total of 200 µL 0.1%TFA solution was added for resolution, and the salt was removed with 80 µg C18. Freeze dried.

## LC-MS analysis

The LC-MS analysis were performed as we previously reported (*Dong et al., 2023*). Briefly, mass spectrometry was performed on Q Exactive HF (ThermoFisher, Waltham, MA, USA) column C18, 3 µm, 150 mm × 75 µm (Eksigent, Dublin, CA, USA).

Mass spectrometry: positive ion detection mode, primary resolution of 120,000, AGC set to 3e6, scanning range of 300–1,400 m/z, maximum injection time was 80 ms, the range of charge scanning was 1–7. Twenty ions with the highest intensity were selected from one MS spectrum for MS/MS analysis. The secondary resolution was 15,000, the AGC was set to 5e4, maximum injection time was 19 ms, collision energy was 30, dynamic exclusion time was 12, and the separation window was 1.6 m/z.

Liquid method: the chromatographic column was C18, 3 µm, 250 mm ×75 µm (Eksigent, Dublin, CA, USA), and the A phase was water, 0.1% formic acid, Phase B consisted of acetonitrile and 0.1% formic acid at a flow rate of 600 nl/min with 40 °C column temperature. Electrospray voltage was 2 kV. The sample volume was 5 µl and the chromatographic gradient was 78 min.

## Peptide identification

The mass spectrometry peak area of peptides is used for quantitative research and statistical analysis. Calculate the multiple changes and $p$-values of peptides in different treatment groups using Matlab's T test algorithm. The selection criteria for differential peptides are based on a foldchange >2, and $p < 0.05$.

## Western blot

Western blot analysis was performed as we previously reported (*Dong et al., 2022*; *Zhao et al., 2023a*). Briefly, human dura mater was lysed with Radio Immunoprecipitation Assay (RIPA) lysis buffer (Beyotime, Shanghai, China) which containing 1mM PMSF (Beyotime, Shanghai, China). The protein level was quantified by BCA Protein Assay Kit (Thermo Fisher, Waltham, MA, USA). 10 $\mu$g protein for each sample were used for Western blot. Proteins were separated on 8% to 20% acrylamide/diacrylamide gels and transferred to PVDF transfer membranes (Millipore, Burlington, MA, USA). The proteins were blocked with 5% BSA for 1 h at room temperature. Then, the PVDF transfer membranes were washed with 1×TBST for three times. Primary antibodies (1:1000-1:2000) were used for membranes incubation at 4 °C for overnight. Then, the membranes were washed with 1×TBST for three times and incubated with second antibodies (1:2500) for 1 h at room temperature. The expression of protein was detected by Tanon 6200 (Tanon, China). Anti-Histone H3 (citrulline R2 + R8 + R17) (H3Cit, ab5103, Abcam, Cambridge, UK), Beta Actin Monoclonal antibody (66009-1-Ig, Proteintech, Chicago, IL, USA) antibody, HPR-conjugated Affinipure Goat Anti-Rabbit IgG (H+L) (SA00001-2, Proteintech, USA) and HPR-conjugated Affinipure Goat Anti-Mouse IgG (H+L) (SA00001-1, Proteintech, Chicago, IL, USA) were used in this study.

## Bioinformatics

The Compute pI/Mw tool (https://web.expasy.org/protparam/) was used to calculate each peptide isoelectric point (pI) and Mw with the default parameters. The KEGG were predicted using KEGG (http://www.kegg.jp/) with the default parameters. The GO pathways were predicted using http://geneontology.org/, and Homo sapiens selected for analysis with the default parameters. The enrichment analysis was performed by the DAVID Functional Annotation Bioinformatics Microarray Analysis Interaction network (https://david.ncifcrf.gov/home.jsp), the default parameters of PValue was used for enrichment analysis. The function of peptide precursor proteins identified by STRING (https://string-db.org, Version 11.0) with the default parameters.

## Statistical analysis

SPSS 20.0 software was used for data analysis, and independent sample $t$-test was used for comparison between the two groups. $p < 0.05$ considered statistically significant.

# RESULTS

## Clinical characteristics of subjects and the research design

Three patients with CSDH and three control patients were recruited for the study. Meninges were collected during surgery. The baseline characteristics of the patients enrolled in this study were listed in Table 1. The experiments design was shown in Fig. 1.

## Meninges peptidomic analyses of CSDH

LC-MS was performed for detected the differentially expressed peptides in the meninges of CSDH compared with relative control. A total of 18,674 peptides were determined in

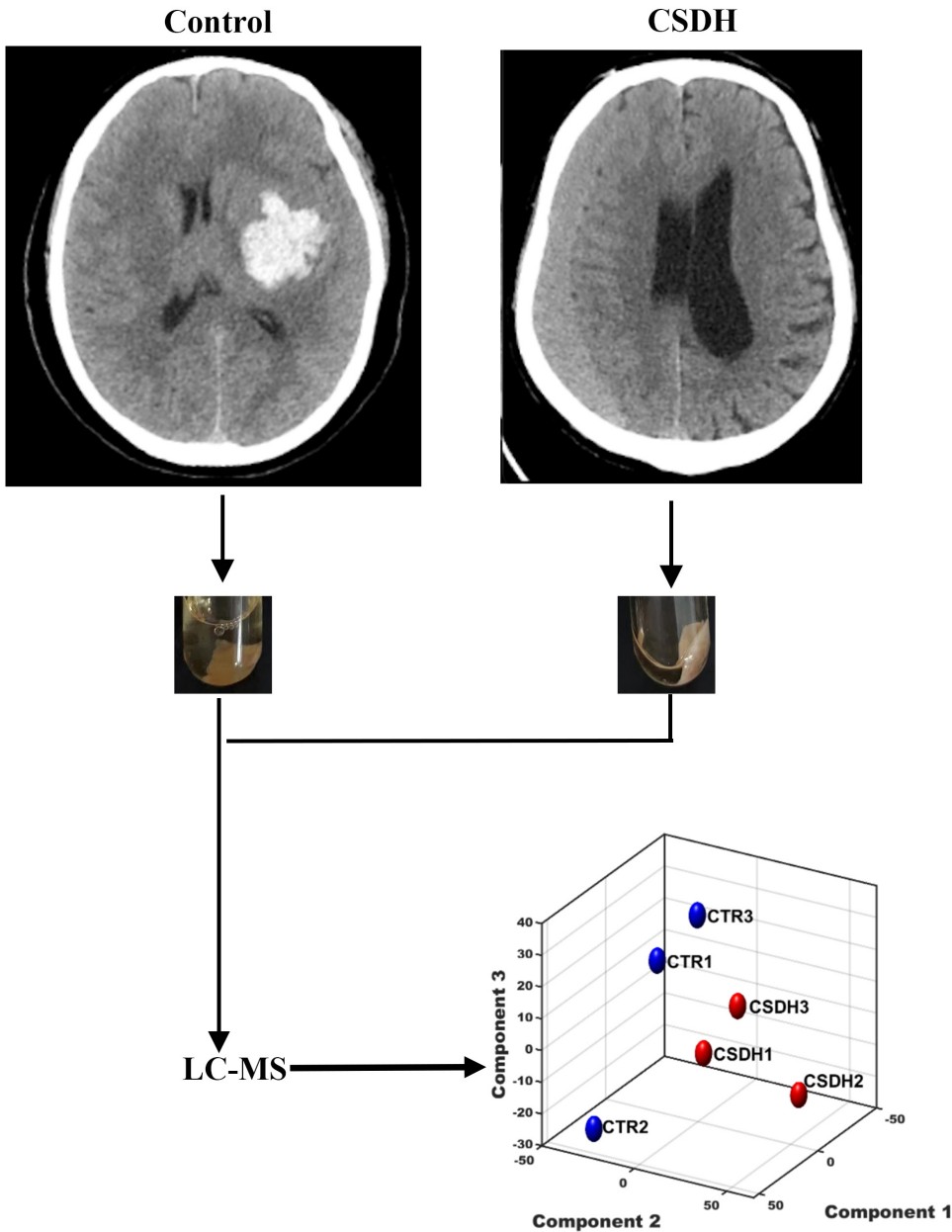

**Figure 1** **The schematic diagram of this study.** Human dura mater was collected from the control and CSDH groups for subsequent LC-MS peptidomics analysis.

human meningeal tissues, of which 569 differential peptides were identified, including 217 up-regulated peptides and 352 down-regulated peptides (Foldchange>2, $p < 0.05$), which were derived from 232 precursor proteins. Hierarchical clustering (Fig. 2A) and volcano mapping (Fig. 2B) displayed the differentially expressed peptides of the CSDH and relative controls. The top 20 up-regulated peptides and the top 20 down-regulated peptide are shown in Tables 2 and 3.

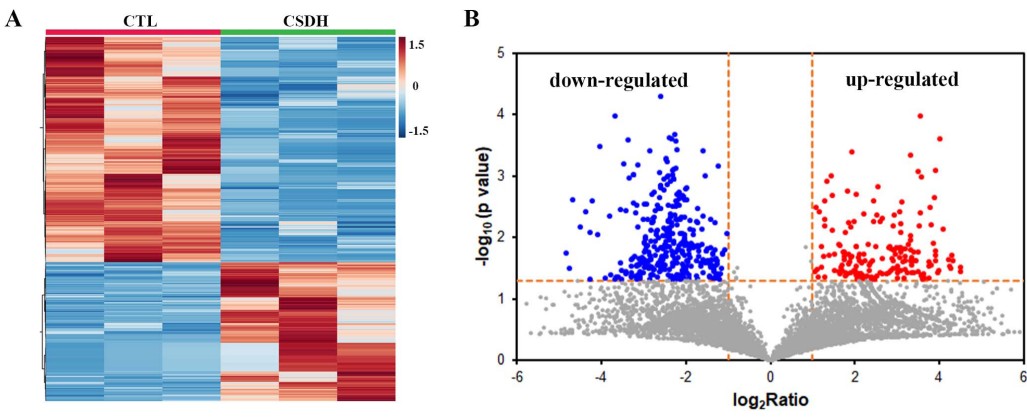

**Figure 2** **Overview of differentially expressed peptides in CSDH compared with control.** (A) Hierarchical map of differentially expressed peptides. (B) Volcano clustering of differentially expressed peptides.

**Table 2** **Top 20 upregulated peptides in CSDH meninges samples compared with control.**

| Peptide | Number | LogRatio | $p$ value | Protein name |
|---|---|---|---|---|
| MNDTVTIRTR | 10 | 7.4680 | 0.0194 | RPS24 |
| TTNIMEILRGDFSSAN | 16 | 4.5223 | 0.0361 | FGA |
| YGGFTVQNEANK | 12 | 4.5106 | 0.0314 | FGB |
| TNIMEILRGDF | 11 | 4.3338 | 0.0246 | FGA |
| GGPGGPGGPGGPMGR | 15 | 4.3138 | 0.0198 | EWS |
| MDSAGQDINLNSPNKGL | 17 | 4.2834 | 0.0326 | TPD52L2 |
| VGGSGGGSFGDNLVT | 15 | 4.2619 | 0.0300 | LMNA |
| SGTAPAAGVVPSR | 13 | 4.2273 | 0.0235 | MAP4 |
| TPTYGDLNHLVS | 12 | 4.1141 | 0.0074 | TBB5 |
| VGGSGGGSFGDNLVTR | 16 | 4.0271 | 0.0002 | LMNA |
| MMLSTEGREGFVVK | 14 | 4.0143 | 0.0350 | HNRH2 |
| LGTRNDLSPTTVM | 13 | 3.9727 | 0.0377 | BLVRB |
| TKGVDEVTIVNILTNR | 16 | 3.9349 | 0.0172 | ANXA2 |
| SPYGGGYGSGGGSGGYGSRRF | 21 | 3.9290 | 0.0008 | ROA3 |
| ALHVPKAPGFAQ | 12 | 3.9275 | 0.0202 | TCPQ |
| VVGVVAGGGRIDKPILK | 17 | 3.9185 | 0.0243 | RPL8 |
| SSKVSRDTLYEAVR | 14 | 3.8906 | 0.0023 | RPL10A |
| RIINEPTAAAIAY | 13 | 3.8764 | 0.0128 | BIP |
| SGNYATVISHNPETK | 15 | 3.8208 | 0.0293 | RPL8 |
| THSLGGGTGSGMGTLL | 16 | 3.8177 | 0.0064 | TUBB5 |

## Characterization of differentially expressed peptides

The general features of the differentially expressed peptides between CSDH and relative control were analyzed. The molecular weights (Mw) of most differentially expressed peptides ranged from 0.8 kD to 2.8 kD (Figs. 3A–3B). The distribution of pI ranged from 3.0 to 7.0 and 8.0 to 12.0 (Figs. 3C–3D).

**Table 3  Top 20 downregulated peptides in CSDH meninges samples compared with control.**

| Peptide | Number | LogRatio | *p* value | Protein name |
|---|---|---|---|---|
| HVQPQPQPKPQVQLHVQSQT | 20 | −5.2108 | 1.3164E−06 | ZYX |
| FAEEGKKLVAASQAALGL | 18 | −4.8221 | 0.0183 | ALBU |
| STRTYSLGSALRPSTSRS | 18 | −4.7418 | 0.0328 | VIM |
| LITKAVAASKER | 12 | −4.6725 | 0.0025 | H1 |
| SALRPSTSRS | 10 | −4.5042 | 0.0068 | VIM |
| GLGAAEFGGAAGNVEAPGETFAQR | 24 | −4.3633 | 0.0038 | GL1AD |
| SVMHEALHNHYTQKSLSLSPG | 21 | −4.2625 | 0.0084 | IGG1 |
| SLNLRETNLDSLPLVDTHSKRTL | 23 | −4.2565 | 0.0495 | VIM |
| SSLNLRETN | 9 | −4.2205 | 0.0026 | VIM |
| AQKQPAGKVQIVSK | 14 | −4.0690 | 0.0093 | MAP4 |
| LQDSVDFS | 8 | −4.0350 | 0.0003 | VIM |
| NLDSLPLVD | 9 | −3.9024 | 0.0463 | VIM |
| YLDHNALESVPLNLPESL | 18 | −3.7876 | 0.0045 | MIME |
| SSVPGVRLLQDSVDFSLADAINT | 23 | −3.7849 | 0.0423 | VIM |
| MELERPGGNEITR | 13 | −3.6801 | 0.0001 | FGA |
| NQASDTFSGIGKKFGLLK | 18 | −3.6601 | 0.0485 | ADIRF |
| NLDSLPLVDTHSKRTL | 16 | −3.6462 | 0.0453 | VIM |
| NIKAPKISMPDLDLNLK | 17 | −3.6007 | 0.0447 | AHNK |
| SLNLRETN | 8 | −3.5334 | 0.0036 | VIM |
| LYASSPGGVY | 10 | −3.5300 | 0.0265 | VIM |

## Gene Ontology and KEGG pathway analysis

GO and KEGG pathway analyses were performed to investigate the function of the differentially expressed peptides. The main biological processes, cellular components and molecular functions enriched by these different peptides are shown in Fig. 4. In the enrichment analysis of biological processes, cytoplasmic translation, translation, DNA replication-independent nucleosome assembly, telomere organization were the most highly enriched subunits (Fig. 4A). In the enrichment analysis of cell components, extracellular exosome, cytosolic ribosome, focal adhesion, cytosolic large ribosomal subunit were the most enriched subunits (Fig. 4B). In the enrichment analysis of molecular functions, RNA binding, structural constituent of ribosome, structural constituent of chromatin, cadherin binding were the most enriched subunits (Fig. 4C). KEGG pathway analysis suggested that the ribosome, neutrophil extracellular trap (NETs) formation, complement and coagulation cascades, spliceosome, platelet activation participate in the pathogenesis of CSDH (Fig. 4D).

## PPI network analysis

Protein interaction network analysis was performed using STRING to detect potential interactions between differentially expressed peptide related precursor proteins. The results shown that 217 proteins were related to each other (Fig. 5).
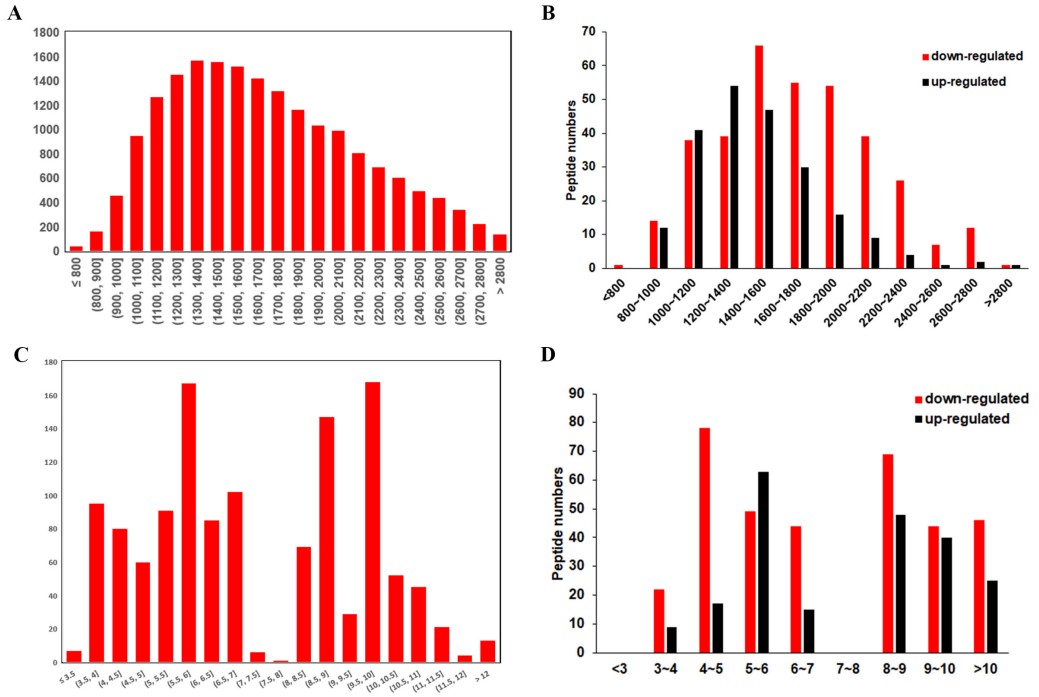

**Figure 3  Characteristic of the differentially expressed peptides.** (A) Molecular weights of the total identified peptides. (B) Molecular weights distribution of the up-regulated and down-regulated peptides. (C) Isoelectric points of the total identified peptides. (D) Isoelectric points distribution of the up-regulated and down-regulated peptides.

## NETs were highly abundant in dura mater and hematoma membrane of CSDH

Inflammation has been reported to play an important role in CSDH, KEGG analysis of the precursor protein of differentially expressed peptides showed that NETs were significantly enriched in CSDH compared with non-CSDH. To determine whether NETs was involved in CSDH, we examined the expression of H3Cit, a NET marker, in human dura and hematoma membrane of CSDH. The Western blot results shown that the protein level of H3Cit were significantly elevated in human dura and hematoma membrane of CSDH compared with in human dura of non-CSDH (6.15 ± 1.23 and 6.10 ± 0.91 *vs* 1.69 ± 0.80) (Fig. 6). This data suggested that NETs might be involved in the pathogenesis of CSDH.

## DISCUSSION

In the present study, we analyzed and identified the expression of peptides in human meninges and compared the differential expression of peptides in CSDH and relative controls. The study is the first to determine the potential function of endogenous peptides may participate in CSDH progress.

Based on the peptidoomics analysis, we demonstrated that a total of 217 up-regulated peptides and 352 down-regulated peptides were differentially expressed in the human dura in the CSDH group compared with the relative control group by LC-MS.

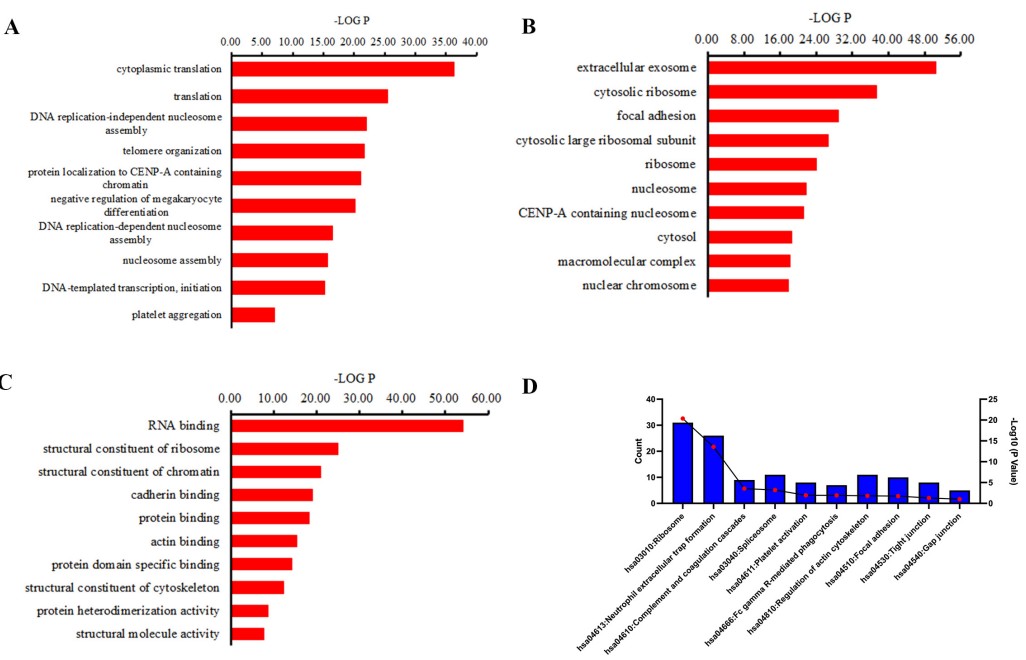

**Figure 4  GO term and KEGG pathway analysis of precursor proteins of differentially expressed peptides.** (A) Biological processes in which precursor proteins of differentially expressed peptides are involved. (B) Cellular components in which precursor proteins of differentially expressed peptides are involved. (C) Molecular function in which precursor proteins of differentially expressed peptides are involved. (D) KEGG analysis of the precursor proteins of differentially expressed peptides.

Most notably, 10 differentially expressed peptides of the top20 down-regulated peptides in CSDH group were derived from VIM. VIM belong to the type III intermediate filament protein, and play an important role in regulates fibrosis, inflammatory response, vascular endothelium, and involvement in neurological diseases (*Ridge et al., 2022*; *Chen et al., 2023*). KEGG analysis revealed that the focal adhesion, tight junction, and gap junction were the dominant enriched pathway in CSDH, which indicates that the VIM-derived peptides may be participate in the CSDH pathological processes through regulation the endothelial cell connection. In addition, fibrinogen alpha chain (FGA) derived peptide TTNIMEILRGDFSSAN and TNIMEILRGDF, and the fibrinogen beta chain (FGB) generated peptide YGGFTVQNEANK were dramatically upregulated in CSDH. Fibrinogen is a crucial blood coagulation protein that is deposited in the brain in many neurological diseases, such as traumatic CNS injury, Alzheimer's disease, and multiple sclerosis (*Bijak et al., 2019*; *Golanov et al., 2019*). The plasma protein fibrinogen enters the CNS through the disrupted blood–brain barrier (BBB) and play an important role in inflammation, coagulation, and tissue repair (*Petersen, Ryu & Akassoglou, 2018*). Fibrinogen derived peptides highly expressed in human dura suggested that these peptides may be involved in the pathological process or may server as the biomarker for CSDH. However, whether the VIM- derived peptides and fibrinogen derived peptide involved in the pathology of CSDH need to be further investigated.

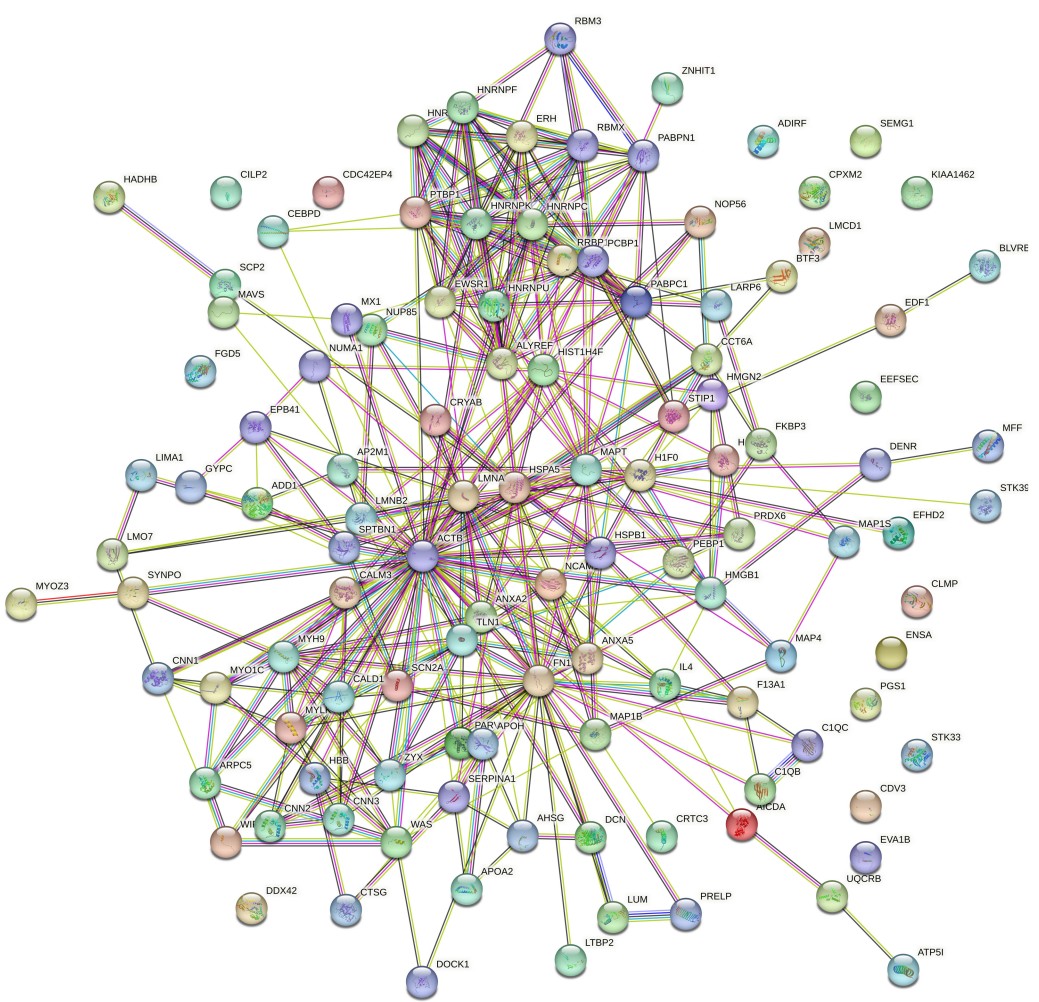

**Figure 5** Protein interaction network analysis of the precursor proteins of differentially expressed peptides by STRING.

Accumulated evidence demonstrated that inflammation play an important role in the development and progression of CSDH (*Edlmann et al., 2017*; *Svedung Wettervik, Sundblom & Ronne-Engstrom, 2023*). Anti-inflammatory drug therapy can dramatically reduce the growth and recurrence of CSDH in conservative and surgically treated patients with CSDH and was considered has potentially beneficial for CSDH patients (*Vychopen, Guresir & Wach, 2022*). Inflammatory cells including neutrophils, macrophages, lymphocytes and eosinophils were participate in the pathogenesis of CSDH. Neutrophils, as a kind of innate immune response effect factor, is produced by bone marrow progenitor cells, is considered have multi-function in regulate many processes, which including autoimmunity, acute injury and repair, and chronic inflammatory processes (*Liew & Kubes, 2019*; *Easton, 2013*). It was reported that the postoperative neutrophil-to-lymphocyte ratio ≥1 was associated with recurrence (*de Oliveira et al., 2022*). In addition, *Li et al. (2014)* found that treatment with the beneficial anti-inflammatory drug atorvastatin in SDHs rats

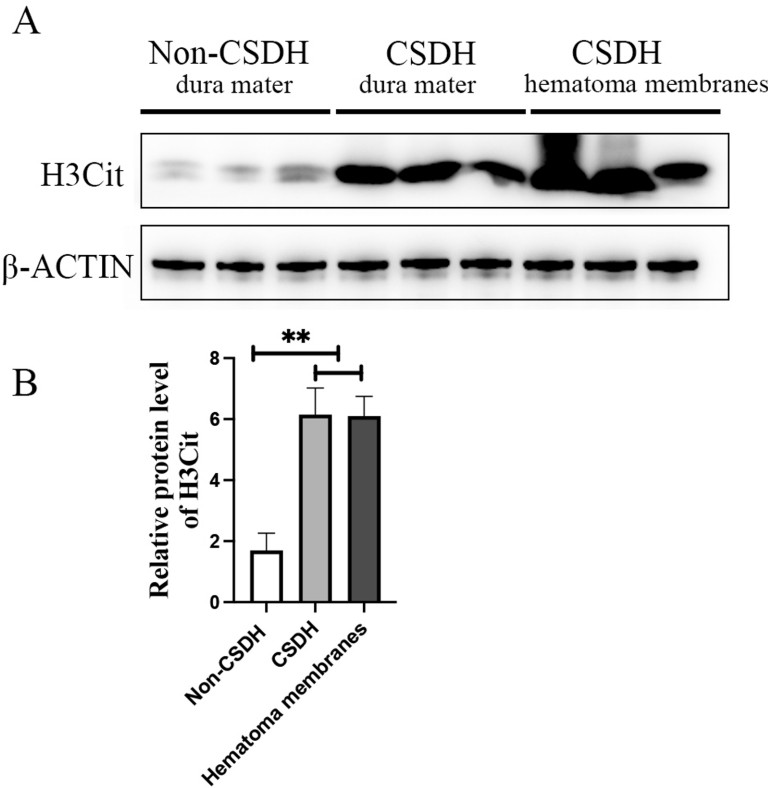

**Figure 6** **H3Cit protein level increased dramatically in the dura mater and hematoma membranes of CSDH compared control.** (A) Western blot analysis of the expression of H3Cit in human dura mater of Non-CSDH, human dura mater of CSDH, and hematoma membranes of CSDH, respectively. (B) The expression of H3Cit which normalized with $\beta$-ACTIN.

significantly reduced the number of neomembranes neutrophils, which indicated the high level of neutrophil might participate in the pathogenesis of CSDH.

Recent studies have shown the presence of a direct ossifying vascular channel connecting the bone marrow of the skull to the meninges, capable of dispersing neutrophils during inflammation (*Cugurra et al., 2021*). The meninges as a myeloid reservoir containing monocytes and neutrophils, however, the role of neutrophils in meninges of CSDH is undetermined. When neutrophils are subjected to various stimuli, they counteract pathogens by producing reactive oxygen species (ROS), proteases, phagocytosis, and neutrophil extracellular traps (NETs) (*Mutua & Gershwin, 2021*; *Castanheira & Kubes, 2019*). Particularly, NETs, which consists of DNA coated with histones, myeloperoxidase, elastase, and cathepsin G (*Brinkmann et al., 2004*), has been reported participate in the intracranial hemorrhage and ischemic stroke (*Zhao et al., 2023b*; *Jin et al., 2022*). However, whether NETs play a role in the pathogenesis of chronic subdural hematoma has not been reported. In this study, we investigated the potential role of peptides derived from human dura, and analyzed KEGG of dysregulated peptide precursor proteins in CSDH patients and found that NETs were enriched in CSDH compared to controls. In addition, Western blot analysis revealed that H3Cit was dramatically increased in the human dura

and hematoma membrane of CSDH, which indicated that NETs formation could be a biomarker of CSDH and NETs are highly related to the development of the CSDH. This result suggested that blockage the NETs generation might have benefit role for therapy of CSDH. However, whether NETs involved in the pathogenesis of CSDH, or whether NETs are therapeutic targets for CSDH remains to be further investigated.

## CONCLUSIONS

In summary, our findings indicated that a total of 569 differentially expressed peptides persist in the dura of CSDH patients. VIM-derived peptides and fibrinogen-derived peptides were dramatically dsyregulated in CSDH compared to without CSDH, indicating the potentially role of VIM, VIM-derived peptides, fibrinogen, or fibrinogen-derived peptides in the pathogenesis of CSDH. Most importantly, we identified that the NETs were remarkably accumulated in the dura and hematoma membranes of the CSDH. These results further understand the role of the meninges in CSDH disease and provide new directions for the treatment of CSDH. However, in this study, only three samples were used in each group for peptidomics analysis, further studies with larger cohorts are needed to identify and validate differentially expressed peptides.

### Funding

This work was supported by the Scientific Research Project of Shanghai Changning District Health Committee under Grant no. 20214Y014 and National Natural Science Foundation of China under Grant no. 82101621. The funders had no role in study design, data collection and analysis, decision to publish, or preparation of the manuscript.

### Grant Disclosures

The following grant information was disclosed by the authors:
Shanghai Changning District Health Committee: 20214Y014.
National Natural Science Foundation of China: 82101621.

### Competing Interests

The authors declare there are no competing interests.

### Author Contributions

- Jie Li conceived and designed the experiments, analyzed the data, prepared figures and/or tables, and approved the final draft.
- Jing Zhao performed the experiments, analyzed the data, prepared figures and/or tables, and approved the final draft.
- Shuchen Sun performed the experiments, authored or reviewed drafts of the article, and approved the final draft.
- Sen Shen performed the experiments, authored or reviewed drafts of the article, and approved the final draft.

- Bincheng Zhong performed the experiments, prepared figures and/or tables, and approved the final draft.
- Xiaohua Dong conceived and designed the experiments, prepared figures and/or tables, authored or reviewed drafts of the article, and approved the final draft.

### Human Ethics

The following information was supplied relating to ethical approvals (i.e., approving body and any reference numbers):

The study was approved by the Medical Ethics Committee of Shanghai Tongren hospital [approval no. 2022-078-01].

### DNA Deposition

The following information was supplied regarding the deposition of DNA sequences:

The mass spectrometry proteomics data are available at the ProteomeXchange Consortium (http://proteomecentral.proteomexchange.org) via the iProX partner repository: PXD046368.

### Data Availability

The raw measurements are available in the Supplementary Files.

### Supplemental Information

Supplemental information for this article can be found online at http://dx.doi.org/10.7717/peerj.16676#supplemental-information.

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
