# Peer review of "Peptidomics insights: neutrophil extracellular traps (NETs) related to the chronic subdural hemorrhage"

_PeerJ, doi:10.7717/peerj.16676_

## Round 0.1 · original submission · Major Revisions

Although one of the reviewers recommended rejection, I decided to give you an opportunity to reply to the critiques and to revise the manuscript.

Reviewer 1 ·

Basic reporting

no comment

Experimental design

no comment

Validity of the findings

no comment

Additional comments

The paper explores the role of endogenous peptides in the pathogenesis of chronic subdural hemorrhage (CSDH). The authors conducted peptidomic profiling of human dura mater samples from CSDH patients and identified differentially expressed peptides. They observed a significant down-regulation of VIM-derived peptides in CSDH compared to those without CSDH, suggesting a potential role for VIM or VIM-derived peptides in the pathogenesis of CSDH. Additionally, the study revealed a substantial enrichment of NETs in CSDH, with elevated levels of H3Cit, a marker of NETs, in the dura and hematoma membranes of CSDH patients. These findings offer fresh insights into the involvement of peptides and NETs in CSDH.

This paper exhibits several strengths:

1. Innovative Methodology: The application of peptidomic profiling to identify differentially expressed peptides in CSDH represents a novel and pioneering approach that enhances our comprehension of the disease's pathogenesis.
2. Profound Findings: The identification of NETs as a significant player in CSDH provides valuable insights into the underlying disease mechanisms.
3. Well-structured Presentation: The paper is effectively organized and presents research findings in a clear, concise manner.

However, several questions remain for the authors to address:

1. The study's sample size is relatively small, potentially limiting the generalizability of the findings. Have the authors considered the feasibility of expanding the sample size to enhance the study's statistical power? While we acknowledge the challenges in sample collection, it would be beneficial if the authors could provide additional reasoning or references to support the adequacy of the sample size for this field.

2. Could the authors provide more comprehensive details regarding the methods employed for the identification and validation of the differentially expressed peptides?

3. The study primarily focuses on identifying differentially expressed peptides and lacks functional experiments to validate the role of NETs in CSDH. Have the authors considered including functional experiments to further investigate the role of NETs in CSDH in future research?

Furthermore, the paper contains some grammatical errors and suggestions for improvement:

Abstract:
1. Replace "and is more common in the elderly" with "and is more common among the elderly."
2. Replace "import" with "important" in "play an import role."
3. Change "is remain" to "remains."
4. Replace "peptidomic profiles was performed" with "peptidomic profiles were performed."
5. In the sentence beginning with "GO (Gene ontology) analysis," rephrase it as "Gene ontology (GO) analysis."

Introduction:
1. In "CSDH is a complex and common neurosurgical disease, which is refers," remove "is" after "which."
2. In "intracranial pressure control," add "the" before "intracranial."
3. In "avoiding multiple surgeries for CSDH," consider rephrasing it as "to avoid subjecting patients to multiple surgeries for CSDH."

Materials & Methods:
1. Change "tissue were collected" to "tissues were collected."
2. Replace "icytoacetic acid" with the correct acid used.

Results:
1. In "differential expressed peptides," change "differential" to "differentially."

Conclusions:
1. In "without CSDH," insert "to" before "without."

Upon addressing these questions and correcting the grammatical errors, this paper should meet the requirements for publication.

Reviewer 2 ·

Basic reporting

Overall Evaluation
Overall, the study aims to explore the role of peptides and Neutrophil Extracellular Traps (NETs) in chronic subdural hemorrhage (CSDH). The topic is of clinical importance and could advance our understanding of the pathogenesis of CSDH. However, there are several concerns, including issues with methodology, statistical robustness, and clarity of presentation that should be addressed before the manuscript is suitable for publication. I recommend major revisions.

Specific Comments
Title and Abstract
The abstract is generally clear but could benefit from clarification regarding the sample size and the study's main contribution. It would help to emphasize what new information this research provides about CSDH and NETs.

Methods
Sample Collection
The sample size is notably small with only three samples for each group. The power of the statistical tests may be compromised, and the results could be less generalizable.
It would be useful to provide more details about how subjects were matched in the control and CSDH groups to control for confounding variables.

Peptide Extraction and LC-MS Analysis
The methods for peptide extraction and LC-MS analysis are generally well described but could benefit from further clarification regarding why specific settings and reagents were chosen.

Results
NETs Analysis
The western blot data showing elevated levels of H3Cit is interesting. However, what is the biological significance of this? Is this elevation in H3Cit specifically due to NET formation or could it also be a marker for something else?

The paper has the potential to make a valuable contribution to our understanding of CSDH. However, I recommend major revisions focusing on improving the statistical rigor, clarifying the methodology, and enhancing the presentation for greater clarity and impact.

Experimental design

No Comments.

Validity of the findings

The sample size is notably small with only three samples for each group. The power of the statistical tests may be compromised, and the results could be less generalizable.

Additional comments

No Comments.

---

## Round 0.2 · Major Revisions

Please address concerns of the reviewer #2 and amend the manuscript accordingly.

Reviewer 1 ·

Basic reporting

No comment

Experimental design

No comment

Validity of the findings

No comment

Additional comments

The authors have effectively addressed all of my queries and implemented the required changes in the manuscript. I therefore recommend accepting this paper.

Reviewer 2 ·

Basic reporting

Abstract:

The abstract presents a concise summary of the study. However, it would be beneficial if the clinical significance or implications of these findings were briefly mentioned.
A brief mention of the potential therapeutic implications or future directions based on these findings could be included at the end.

Experimental design

Materials & Methods:

Sample Collection:
It is commendable that the authors declared obtaining written informed consent from participants. However, there should be a bit more clarity on how patients were chosen for each group, especially given the small sample size.

LC-MS Analysis:
It might be helpful to include any specific settings or details on data analysis methods, especially for readers unfamiliar with the technique.

Bioinformatics:
The tools and databases used are appropriately cited. However, any thresholds or specific parameters used in the analysis should be stated.

Results:

Clinical characteristics of subjects and the research design:
The baseline characteristics should be presented in a clear manner in Table 1. Given the small sample size, it might be important to address the statistical power and potential biases in discussioin.

Meninges peptidomic analyses of CSDH:
The results are presented in a comprehensive manner. However, further discussion on the potential implications or significance of the top differentially expressed peptides might be beneficial.


NETs were highly abundant in dura mater and hematoma membrane of CSDH:
While the abstract mentions H3Cit elevation, the results section cuts off before explaining these findings in detail. The actual results for H3Cit expression should be presented, possibly with quantitative data.
General Comments:

The use of only three samples in each group raises concerns about the statistical power and potential for overfitting. The authors should address these concerns, possibly suggesting that this is a preliminary study and that further research with larger cohorts is needed.

It would be beneficial if the authors discuss the potential clinical implications or therapeutic interventions based on their findings in the discussion section.

Overall, the manuscript presents interesting findings that could have implications in the understanding and management of CSDH. However, further validation with larger samples and a more detailed discussion of the results in the context of existing literature would enhance the paper's impact.

Validity of the findings

No Comments.

Additional comments

No Comments.

---

## Round 0.3 · accepted · Accept

All concerns were adequately addressed and the manuscript was amended accordingly. Therefore, revised version is acceptable now.

Reviewer 2 ·

Basic reporting

The authors have addressed all my concerns.

Experimental design

The authors have addressed all my concerns.

Validity of the findings

The authors have addressed all my concerns.